# Ladder Use Ability, Behavior and Exposure by Age and Gender

**DOI:** 10.3390/geriatrics9030061

**Published:** 2024-05-10

**Authors:** Erika M. Pliner, Daina L. Sturnieks, Kurt E. Beschorner, Mark S. Redfern, Stephen R. Lord

**Affiliations:** 1Department of Mechanical Engineering, University of Utah, Salt Lake City, UT 84112, USA; 2Rocky Mountain Center for Occupational and Environmental Health, University of Utah, Salt Lake City, UT 84111, USA; 3Falls, Balance and Injury Research Centre, Neuroscience Research Australia, Randwick, NSW 2031, Australia; d.sturnieks@neura.edu.au (D.L.S.); s.lord@neura.edu.au (S.R.L.); 4School of Biomedical Sciences, University of New South Wales, Sydney, NSW 2052, Australia; 5Department of Bioengineering, University of Pittsburgh, Pittsburgh, PA 15261, USA; beschorn@pitt.edu (K.E.B.); mredfern@pitt.edu (M.S.R.); 6School of Population Health, University of New South Wales, Sydney, NSW 2052, Australia

**Keywords:** ladder falls, aged, risk-taking, task performance, unsafe behavior, injury prevention

## Abstract

This study aimed to quantify and compare ladder use ability and behavior in younger and older men and women from three ladder use behavior experiments. The experimental tasks comprised (1) changing a lightbulb on a household stepladder under two cognitive demands (single and dual task), (2) clearing a simulated roof gutter on a straight ladder and (3) querying ladder choice in different exigency scenarios. Ladder use ability and behavior data were captured from recorded time, performance, motion capture and user choice data. In addition, this study surveyed ladder use frequency and habitual behaviors. The experimental findings indicate that older adults require more time to complete ladder tasks; younger adults display riskier ladder use behaviors; men and women display similar ladder use ability; and men are more willing to climb riskier ladders. The survey found older adults to report more frequent ladder use than younger adults, and men use straight ladders more frequently than women. These results suggest that the reported higher ladder fall rates experienced by older adults and men are linked to increased ladder use exposure and riskier ladder choice. This knowledge can help guide population-specific interventions to reduce ladder falls in both young and older people.

## 1. Introduction

Many serious fall injuries across the globe involve a fall from a ladder. The average domestic ladder fall rate in Denmark has been reported to be 0.80/1000 people/year [1], and alarmingly, there have been 50% plus increases in domestic ladder fall rates in Australia over a 10-year period [2] and in the United States over a 16-year period [3]. Many people who suffer ladder falls sustain serious injuries that require multiple-day hospitalizations and lead to persisting disability and even death [1,2,3,4,5,6,7,8,9,10,11]. Older adults have higher ladder fall rates compared to younger adults in both domestic [1,2,6,11] and occupational [12,13,14] settings. Men have higher rates across all ages compared to women [1,2,3,6,9,10,11,12,14,15].

The higher fall rates in older adults and older men, in particular, may relate to ability, behavior and exposure factors. For example, while younger adults experience more ladder climbing slips than older adults [16], they are likely more able to recover from such perturbations, due in part to greater upper body strength [17]. The capability to generate upper body force greater than one’s bodyweight is a critical factor for arresting a ladder fall [18,19,20]. The importance of physical attributes on ladder use ability has also been highlighted among older adults, where greater strength and better balance are associated with faster ladder task completion times [21,22]. In addition, older adults also have more limited cognitive reserves, which may increase the risk of ladder falls, as ladder tasks require cognitive resources for standing stability and task completion [23,24].

In terms of gender differences, women have less strength [25] and poorer balance and functional mobility in challenging conditions compared to men [26]. Men are more likely to be risk-takers than women [27], which in the context of ladder use may manifest as behaviors that challenge stability limits [21,22]. Finally, ladder fall rates may be proportional to ladder use exposure, which may differ between men and women and between younger and older adults.

Experimental ladder safety research has focused primarily on a single age group [16,17,21,22,28,29,30,31,32,33,34,35,36,37,38]. The majority of this research has been conducted on healthy younger adults or occupational ladder users [16,17,28,29,31,33,34,35,36,37,38]. Ladder use is a complex task, requiring physical and cognitive attributes [21,22]. While age is known to affect physical and cognitive ability, there has been limited investigation on ladder use ability and behavior between age groups. Similarly, there is a needed investigation of ladder use ability and behavior between men and women.

Understanding age and gender differences in factors related to ladder fall risk (i.e., ability, behavior and exposure) may assist in devising demographic-specific interventions to prevent ladder falls. Thus, this study assessed ladder fall risk by quantifying ladder use ability and behavior in three simulated household ladder task experiments and surveyed habitual ladder behavior and exposure in younger and older men and women. We hypothesized (registered at https://osf.io/x9gqp; last accessed on 8 May 2024) that (i) older adults would display poorer ladder use ability compared to younger adults, particularly when performing a secondary cognitive task, (ii) women would display poorer ladder use ability compared to men and (iii) older adults and women would demonstrate less risky ladder use behavior compared to younger adults and men, respectively. We also investigated ladder use exposure and general behavioral metrics (i.e., everyday risk-taking and fear of falling) across age groups and genders.

## 2. Materials and Methods

### 2.1. Participants

One hundred and four healthy older adults and 20 healthy younger adults were recruited through flyers in the greater Sydney area, a volunteer research registry list for Neuroscience Research Australia and word of mouth. The older adults were aged 65+ years, lived independently in the community and did not use an assistive device in their homes. The younger adults were aged between 18 and 40 years. Exclusion criteria for both groups were the presence of neurological disorders and an inability to complete the ladder tasks without pain. Ethical approval was obtained from the University of New South Wales Human Research Ethics Committee, and all participants provided written consent to participate in the study. This study differs from previous work on this dataset [21,22,30,32,39] through the inclusion of younger adults, inclusion of a behavioral risk ladder use experiment, analysis across all ladder use experiments and comparison between age group and gender.

### 2.2. Ladder Use, Risk-Taking and Concern about Falling

Participants completed the Unsafe Ladder Use Behavior Scale, which comprised 11 yes or no questions [39] from the Ladder Use History Survey (Appendix A). The sum of unsafe ladder use behaviors (range from 0 to 11) was used as the test measure. Ladder use frequency for different ladder types (step, straight, fixed) was also recorded for each participant. In older participants, concern about falling was assessed using the Iconographical Falls Efficacy Scale, comprising a 10-item survey with a 4-point response option (not at all concerned, somewhat concerned, fairly concerned, very concerned), scaled 1 to 4, where a higher score denoted a greater concern about falling [40]. Risk-taking behavior was assessed using the Everyday Risk-Taking Scale, comprising 10 items with a 4-point response option (never, occasionally, mostly, always), scaled 1 to 4, where a higher score denoted greater everyday risk-taking [41].

### 2.3. Experimental Protocol

Participants completed three laboratory-based ladder use experiments to quantify ladder use ability and behavior.

#### 2.3.1. Changing a Lightbulb on a Household Stepladder

Participants were asked to climb to the second step of a household stepladder to change a lightbulb as a single task while concurrently undertaking a cognitive (naming animals) task (dual task) (Figure 1a), as detailed in Pliner et al. 2021. Participants were instructed to complete the task as quickly and as safely as possible. Ladder use ability was quantified as task completion time, standing stability and animal naming rate. Task completion time was taken as the time taken to climb the stepladder, replace the lightbulb and descend the ladder, with faster times indicating more efficient ladder work. Standing stability was quantified from the center of pressure (COP) elliptical area and the minimum distance between the COP and posterior edge of the ladder step (referred to hereafter as edge distance) calculated from the data of two force plates (sampling at 200 Hz) beneath the ladder (methods detailed in Appendix B). Smaller elliptical areas (more controlled balance) and greater edge distance (reduced backward balance loss risk) were considered to indicate better standing stability. The animal naming rate was quantified by the number of animals named during the dual task, normalized to the task completion time.

#### 2.3.2. Clearing a Simulated Roof Gutter on a Straight Ladder

Participants were asked to climb to the third step of a straight ladder to clear a simulated roof gutter (Figure 1b), as described in Pliner et al. 2020. The starting location of the straight ladder was positioned in the middle of a horizontal gutter, which was 5.8 m in length and completely filled with tennis balls. Thus, participants were unable to clear the gutter in one ladder climb and needed to descend, move and reclimb the ladder as often as needed to clear the gutter. Participants were instructed to complete this task as quickly and as safely as possible. Ladder use ability was quantified as task completion time and the number of ladder moves taken to clear the gutter, where a faster time and fewer ladder moves indicated more efficient ladder work. Risky ladder use was quantified from the maximum reach (normalized to arm span) and lean distance (normalized to height) from the ladder center, with farther reach and lean distances indicating riskier ladder use behavior. Participant kinematics were captured from 51 reflective markers that were placed on anatomical landmarks and recorded using an 8-camera motion capture system (sampling at 100 Hz) (Vicon Motion Systems Ltd., Oxford, UK). Reach and lean distances were calculated from markers placed on the distal end of the 2nd metacarpals bilaterally, 7th cervical vertebra, anterior superior iliac spines and posterior superior iliac spines, relative to six reflective markers placed on the straight ladder (three markers per ladder rail at the height of the 1st–3rd rungs) (methods detailed in Appendix C).

#### 2.3.3. Ladder Choice in Different Exigency Scenarios

In this experiment, participants were asked if they would climb a series of ladders under differing exigency scenarios (Figure 1c). The scenarios involved climbing a ladder to (Figure 1a) wash a window and (Figure 1b) push an emergency escape button. The ladders ranged in tipping risk due to their design dimensions (i.e., height and width) and were arranged from the most to least stable (least to most risky) ladder to climb (numbered 1 to 8 in Figure 1c). The first “ladder” comprised a one-step block (10 cm × 32 cm × 32 cm) and the second “ladder” comprised a two-step block (30 cm × 32 cm × 32 cm). The next three ladders had a base-of-support width of 47 cm with lengths of 183 cm, 274 cm and 366 cm. The sixth and seventh ladders had the same length of 366 cm with base-of-support widths of 37 cm and 26 cm, respectively. Ladders 3 to 7 had two rails supporting rungs every 31 cm. The final ladder had a length of 366 cm and a base-of-support width of 3 cm, i.e., a pole–peg ladder with rungs alternating from each side of a single rail every 15 cm. The straight ladders (ladders 3 to 8) were inclined at the recommended ladder setup angle of 75.5 degrees from the horizontal [28,29,42]. Simulated emergency escape buttons were placed directly above each ladder, and simulated windows were placed above and to the right of each ladder (depicted in Figure 1c).

First, participants were asked if they would climb the one-step block to push the emergency escape button to avoid waiting in an abandoned warehouse for six hours. Participants were then asked the same question in relation to climbing the remaining ladders in the order from the next minimal risk ladder (two-step block) to the riskiest ladder (pole–peg ladder). If the participant said ‘No’ to climbing a ladder, they were asked if they would climb the ladder to avoid waiting in an abandoned warehouse overnight. The riskiest ladders the participant indicated they would climb to avoid a six-hour and overnight wait in an abandoned warehouse were recorded.

Participants were then given a washcloth and asked to step onto the one-step block and wash the simulated window. Next, with the washcloth still in hand, participants were guided along the ladders (least risky to most risky) and asked if they would climb the ladder to wash the window in the laboratory today. If they said ‘No’ to climbing a ladder, they were not asked if they would climb the remaining riskier ladders. Starting from the riskiest ladder the participant was willing to climb in the laboratory to wash the window, they were asked if they would also climb that ladder at home to wash a window, from most risky (i.e., ladder they were willing to climb in the laboratory to wash the window) to least risky ladder until they said ‘Yes’. The riskiest ladders the participant indicated they would climb to wash a window in the laboratory and at their home were recorded. The participants did not climb any of the ladders (except the one-step block), but the questions were presented as if they would be climbing the ladders during the experiment. Younger adults were not asked the riskiest ladder they would climb to wash a window at home.

### 2.4. Statistical Analysis

Logarithmic (task completion times, elliptical area and concern about falling) or square root (maximum lean) transformations were first performed to correct positively-skewed distributions and allow parametric analysis. Three-way ANOVAs were performed on task completion time, elliptical area and edge distance from the lightbulb changing task with age group, gender, cognitive demand and first- and second-order interactions as factors. Two-way ANOVAs were performed on the animal naming rate in the lightbulb changing task, maximum reach, maximum lean, task completion time and number of ladder moves in the gutter clearing task, surveyed unsafe ladder behaviors and everyday risk-taking with age, gender and first-order interaction as factors. If significant interactions were found, Fisher’s LSD post hoc comparisons were performed to determine which groups significantly differed. An independent t-test was performed to examine concerns about falling between the older men and women.

Ordinal logistical regression analyses were performed on the riskiest ladder choice and ladder use frequency by ladder type (step, straight, fixed) with age group and gender as predictors. Ordinal variables for the riskiest ladder choice comprised the eight ladder choice options described above, arranged from least risky (one-step block) to riskiest (pole–peg ladder) ladder to climb. Ordinal variables for ladder use frequency comprised six ladder use rates (Never, Once a Year, Few Times a Year, Monthly, Weekly, Few times a Week), arranged from no use (Never) to high-frequency use (Few times a Week). The exigency scenario was an additional predictor added to the model of the riskiest ladder choice. If significant interactions were found, odds ratios were reported to determine which groups significantly differed. Statistical software (IBM SPSS, Version 24. IBM Corp., Armonk, NY, USA) was used to perform the analyses, and significance levels were set at *p* < 0.05.

## 3. Results

The older participants (*n* = 104) included 52 (50%) women and the younger participants (*n* = 20) included 10 (50%) women (Table 1). 

### 3.1. Ladder Use, Risk-Taking and Concern about Falling

Ladder use and type varied across both age group and gender (Table 2). Older adults reported more frequent stepladder use (age group main effect: W_1_ = 7.96; *p* = 0.005) but less frequent fixed ladder use (age group main effect: W_1_ = 18.31; *p* < 0.001) than younger adults. Men used straight ladders more frequently than women (gender main effect: W_1_ = 7.14; *p* = 0.008) but reported similar stepladder (W_1_ = 2.80; *p* = 0.094) and fixed ladder (W_1_ = 1.83; *p* = 0.176) use with women. Younger adults reported more unsafe ladder behaviors (age group main effect: F_1,117_ = 11.48; *p* = 0.001) and greater everyday risk-taking than older adults (age group main effect: F_1,120_ = 32.03; *p* < 0.001). Men and women reported a similar number of unsafe ladder behaviors (F_1,117_ = 2.18; *p* = 0.143), and older women reported greater concern about falling than older men (gender main effect: F_1,102_ = 6.87; *p* = 0.010). There were no interaction effects between age and gender on ladder use frequency, unsafe ladder behaviors and everyday risk-taking (Table 2).

### 3.2. Changing a Lightbulb on a Stepladder

Younger adults had faster task completion times (age group main effect: F_1,119_ = 22.35; *p* < 0.001) and a higher animal naming rate in the dual task condition (age group main effect: F_1,119_ = 23.80; *p* < 0.001) than the older adults (Table 3). Cognitive demand (single vs. dual task) had a significant effect on task completion time (F_1,119_ = 12.62; *p* < 0.001) with both age groups being slower in completing the lightbulb task when naming animals (dual task condition) compared to the single-task condition. Regarding gender comparisons, the men and women had similar task completion times (F_1,119_ = 2.68; *p* = 0.104) and had comparable animal naming rates in the dual task condition (F_1,119_ = 0.54; *p* = 0.464). There were no interaction effects between age group, gender and cognitive demand on task completion time (Table 3 and Table 4).

There were no age group (F_1,111_ = 3.38; *p* = 0.069) or cognitive demand (F_1,111_ = 2.38; *p* = 0.126) main effects for the elliptical area (Table 3 and Table 4). Men, however, had larger elliptical areas than women (gender main effect: F_1,111_ = 11.11; *p* = 0.001), with the younger men displaying high excursions in the dual task condition relative to the other groups (Figure 2c). There were no interaction effects between age group, gender and cognitive demand on the elliptical area (Table 3 and Table 4).

Younger adults achieved greater edge distances (age group main effect: F_1,111_ = 4.88; *p* = 0.029) than older adults (Table 3). There was a significant age group × cognitive demand effect on edge distance (F_1,111_ = 4.97; *p* = 0.028), with the younger adults decreasing their edge distance with additional cognitive loading from the dual task condition compared to the single-task condition (Fisher’s LSD post hoc: F_1,111_ = 4.11; *p* = 0.045), whereas the edge distance of older adults did not vary with cognitive demand (Fischer’s LSD post hoc: F_1,111_ = 0.86; *p* = 0.355) (Figure 2d). There was no difference in edge distance between the men and women (F_1,111_ = 0.11; *p* = 0.739). There were no other interaction effects between age group, gender and cognitive demand on edge distance (Table 3 and Table 4).

### 3.3. Clearing a Simulated roof Gutter on a Straight Ladder

The younger adults cleared the simulated roof gutter in less time (45.4 s faster) (age group main effect: F_1,112_ = 19.86; *p* < 0.001) with fewer ladder moves (average 0.5 less) (age group main effect: F_1,112_ = 6.27; *p* = 0.014) compared to the older adults (Table 3; Figure 3). Younger adults also reached (age group main effect: F_1,117_ = 6.06; *p* = 0.015) and leaned (age group main effect: F_1,109_ = 4.77; *p* = 0.031) farther than older adults with respect to their arm span and height (Table 2). Men and women recorded similar times for clearing the simulated roof gutter (F_1,112_ = 2.03; *p* = 0.157), took a similar number of ladder moves to complete the task (F_1,112_ = 3.68; *p* = 0.058) and displayed similar reaching (F_1,117_ = 0.55; *p* = 0.458) and leaning (F_1,109_ = 0.36; *p* = 0.552) behavior (Table 2 and Table 3). There were no interaction effects between age group and gender on task completion time, number of ladder moves and maximum reach and lean distances for clearing a roof gutter on a straight ladder (Table 2 and Table 3).

### 3.4. Choosing a Ladder in Different Exigency Scenarios

Ladder choice was influenced by the exigency scenario (exigency scenario main effect: W_3_ = 74.43; *p* < 0.001) (Table 4). Specifically, participants were 20 times more likely to climb a riskier ladder to avoid waiting in an abandoned warehouse overnight than to wash a window at home (odds ratio = 19.9; CI = 11.3–35.2; *p* < 0.001). Younger and older adults chose similar ladders (W_1_ = 1.42; *p* = 0.234) (Table 2), but men chose riskier ladders than women across the exigency scenarios (gender main effect: W_1_ = 5.04; *p* = 0.025) (Table 2). There were no interaction effects between age group, gender and exigency scenario on ladder choice (Table 4).

## 4. Discussion

This study found younger adults to have superior ladder use ability than older adults, with faster task completion times, greater reach, better stability and superior dual task performance. On the other hand, younger adults took greater risks during ladder use and reported more unsafe ladder behaviors compared to older adults. Across the tasks, men and women were found to have similar ladder use ability, but men were willing to climb riskier ladders than women. Surveyed measures complemented these experimental findings, with younger adults reporting greater everyday risk-taking compared to older adults and older women reporting a greater fear of falling than older men. Younger adults reported using fixed ladders more frequently than older adults, older adults reported using stepladders more frequently than younger adults and men reported using straight ladders more frequently than women. This knowledge can guide population-targeted training programs for more effective ladder fall intervention programs. 

Younger adults performed better than older adults with respect to most ladder task time and stability measures. This is likely linked to greater physical and cognitive abilities in younger adults than older adults and agrees with the ladder use literature that has found greater strength and better balance and cognition to be associated with faster ladder task completion times [21,22]. However, the additional cognitive loading (dual task condition) did not exacerbate this difference. In fact, while the dual task resulted in similar increases in task completion times for changing a lightbulb on a household stepladder across younger and older adults, stability worsened during the dual task condition in the younger adults only, partially rejecting our first hypothesis. These dual task findings conflict with those of previous studies that have reported greater reductions in balance and task performance in older adults compared with younger adults when the task demands become more challenging with increased cognitive loading [23] or postural threat [43]. Notably, we found the younger adults had faster animal naming rates than the older adults, suggesting an increased relative prioritization of the secondary task over balance. Thus, our findings agree with the concept of balance prioritization in older adults and secondary task prioritization in younger adults when standing at elevated levels [43].

Our finding that women were no worse than men in completing ladder tasks is contrary to our second hypothesis. There were no gender differences in the task time metrics of changing a lightbulb or clearing a gutter on a ladder. Indeed, men displayed reduced stability in the lightbulb task, indicated by a greater elliptical area, compared to women, which may indicate the men were less cautious or more unstable while completing the task. This difference may have been due to women being more cautious, driven in part by fear, as the older women reported greater concern about falling than the older men, which is known to increase postural rigidity (e.g., increase muscle co-contraction and sway frequency and reduce elliptical area) [44,45,46]. However, these associations are complex, since high levels of muscle co-contraction have also been associated with reduced stability and increased fall risk [47,48], and adaptive postural strategies can vary among subgroups of people (e.g., anxious vs. non-anxious) [49]. Thus, further investigation is warranted on standing stability measures (e.g., muscle co-contraction, sway frequency) to comprehensively understand fall risk in men and women working from a ladder.

We confirmed our third hypothesis that the younger adults reached and leaned farther when clearing the roof gutter on a straight ladder and reported a greater number of unsafe ladder behaviors than the older adults. They did not, however, choose riskier ladders than the older adults across the ladder exigency scenarios. The inconsistency in these findings may relate to the risk context [50,51,52] in that the ladder exigency choice experiment involved trial-by-trial feedback (previously found to be similar between older and younger adults [51]), whereas the clearing the roof gutter task did not. In the ladder choice experiment, participants stood in front of each ladder, allowing them to assess their risk in an unhurried trial-by-trial manner, which may have led to similar ladder choices between age groups. In contrast, in the gutter clearing experiment, the superior physical ability of the younger adults may have allowed them to take greater risks by reaching and leaning farther in a task that emphasized fast completion times as a priority.

Men chose riskier ladders across the ladder exigency scenarios than women, but the women took similar risks during the ladder tasks (i.e., reaching and leaning) and reported a similar number of unsafe ladder behaviors as men, in contrast with our third hypothesis that women would demonstrate less risky ladder use behavior compared to men. The stability of the ladders in the ladder choice (non-secured ladder) and gutter (fixed to apparatus) experiments may have contributed to the varying risk behavior of women in these experiments. Further, men and women recruited in this cohort may have been greater risk-takers than the general population, as eligibility for this study included willingness to climb ladders, which is a risk-inherent everyday task.

Findings from this study assist in explaining inconsistencies between epidemiological reports that show older adults and men experience more ladder falls and experimental studies that show younger adults and women to be at greater ladder fall risk. Younger adults had better ladder use ability and reported less frequent ladder use than older adults when considering all ladder tasks, which would reduce their ladder fall exposure risk compared to older adults. Thus, longer task times and greater ladder use frequency are likely two of the factors contributing to the documented increased ladder fall injury rates in older adults. This is despite younger adults having riskier ladder use behaviors (farther reaching and leaning, self-reported unsafe ladder behaviors) than older adults. Riskier ladder use behaviors can lead to unsafe body positioning with the ladder (e.g., leaning outside the ladder side rails) [53], challenging stability limits and increasing ladder fall risk. Thus, riskier ladder use behaviors in younger adults may contribute to younger adults experiencing more ladder climbing slips than older age groups in experimentally based ladder fall research [16]. However, additional work is needed on ladder perturbation exposure and fall outcomes between age groups. This can be investigated through ladder use perturbation experiments [31] or by recording near-misses and fall outcomes in ladder field research [54]. 

While women have been shown to have reduced performance in arresting a ladder fall in the laboratory [31], we found no indication of poorer ladder use ability or riskier behaviors in women to contribute to this risk. Interestingly, we found men and women reported a similar frequency of stepladder and fixed ladder use in the domestic setting. However, men reported more frequent straight ladder use, which may be linked to their willingness to climb riskier ladders (i.e., ladders with a greater step height and narrower base of support). Further, a large portion of ladder falls have been reported to occur from straight ladders [5,6]. Thus, the discrepancy in ladder falls by gender may be predominantly driven by straight ladder use in men.

We acknowledge a number of study limitations. First, more older adults were recruited than younger adults, and the younger adults showed more heterogeneity in standing stability metrics for stepladder use than older adults. Future studies investigating standing balance parameters in individuals working from heights should include sample sizes larger than 20 participants. Second, the generalization of these findings to other populations warrants investigation. Third, muscle activity was not investigated to complement the standing stability metrics to better understand physiological contributions to ladder fall risk. Fourth, the straight ladder in the simulated gutter clearing experiment was fixed to the experimental apparatus. Thus, ladder use ability and behavior metrics in this task may differ from most real-world scenarios where ladders are not firmly fixed in place. Finally, participants in this study were not subjected to ladder use perturbations, which may provide further insight into ladder fall risk. However, perturbing older adults from a ladder poses participant safety challenges (e.g., high breaking forces from a safety harness) that require resolution prior to testing. Additional work is needed to confirm the association between ladder ability, behavior and exposure metrics with ladder falls as an outcome. Overall, there is a need for future studies that consider greater population variability, causal factors of ladder use behavior and real-world scenarios. 

## 5. Conclusions

The aim of this study was to determine age and gender differences between ladder use ability, behavior and exposure. We (i) confirmed that older adults display poorer ladder use ability than younger adults. We (ii) did not confirm that women display poorer ladder use ability compared to men. We (iii) confirmed that older adults display less risky ladder use behaviors than younger adults and men choose riskier ladders to climb than women, but men and women display similar risk during ladder use. 

This study helps explain some inconsistencies between epidemiology records and experimental studies of ladder falls. We found increased ladder fall risk exposure in older adults due to longer task times and more frequent ladder use compared to younger adults, likely contributing to their high ladder fall injury rates. Younger adults displayed riskier ladder use behaviors than older adults, likely contributing to their higher climbing slip rates that have been previously reported. Furthermore, the previously reported increased ladder fall rate among men compared to women appears to be driven by greater reported use of straight ladders and men being more willing to climb riskier ladders than women. This knowledge may assist in guiding population-specific ladder fall interventions. These should prioritize improving ladder use ability and/or reducing inappropriate ladder use in older adults, focus on reducing risky ladder use behaviors in younger people and be tailored towards straight ladder use safety and the selection of appropriate ladders to accomplish tasks in men.

## Figures and Tables

**Figure 1 geriatrics-09-00061-f001:**
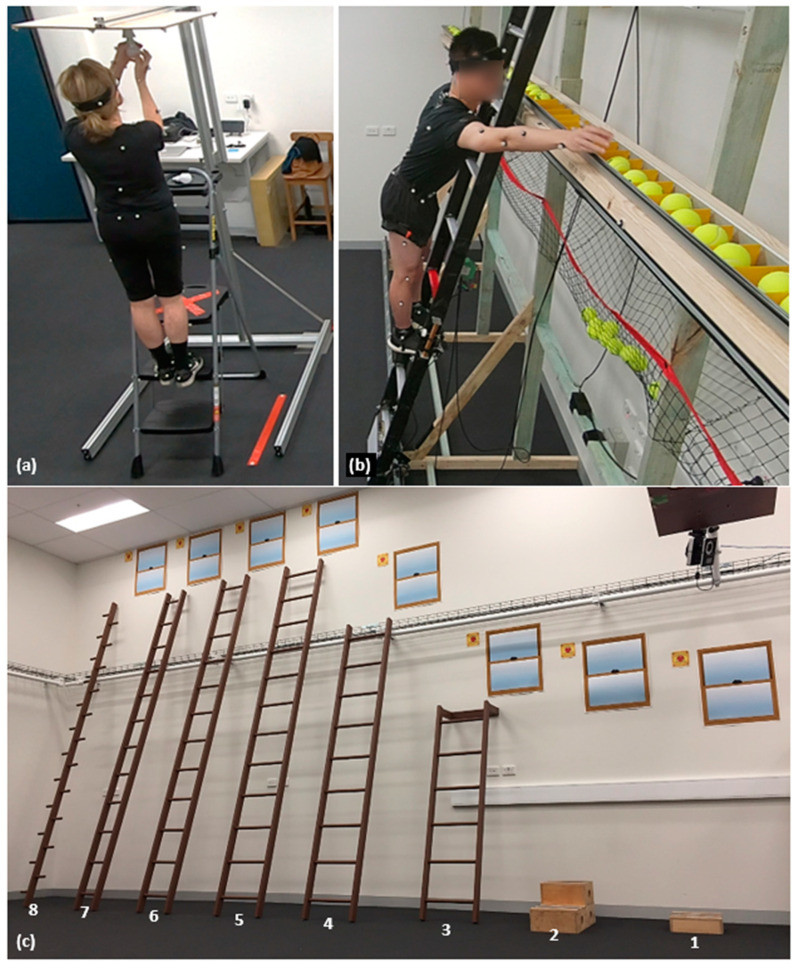
The apparatuses for the three ladder use experiments of changing a lightbulb on a household stepladder (**a**), clearing a simulated roof gutter on a straight ladder (**b**) and ladder choice (ranging from a 1-step box of 4” height to 12 ft. length pole–peg ladder) to wash a window and to push an emergency button to avoid a wait in an abandoned warehouse (**c**).

**Figure 2 geriatrics-09-00061-f002:**
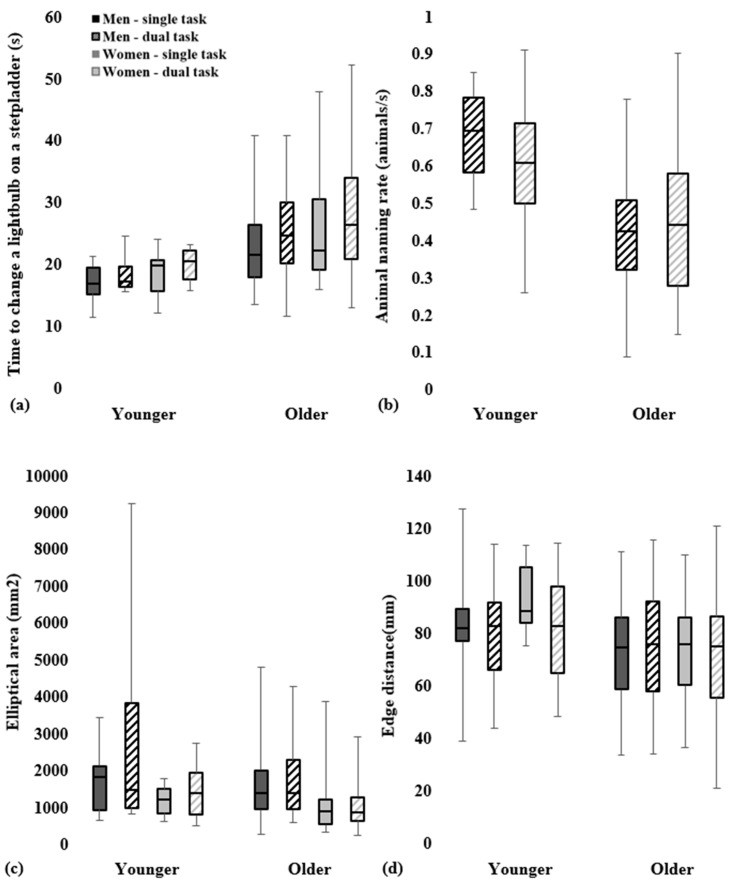
Task performance metrics in changing a lightbulb on a household stepladder. Box plots of the task completion time (**a**), animal naming rate (**b**), elliptical area (**c**) and edge distance (**d**) of older and younger adults. Box plots denote the median (center line), first quartile (lower edge of box), third quartile (top edge of box), minimum (lower whisker) and maximum (upper whisker) values of the depicted groups. Depicted groups comprise gender (men in dark gray, women in light gray) and cognitive demand (single task in solid filled box, dual task in striped filled box).

**Figure 3 geriatrics-09-00061-f003:**
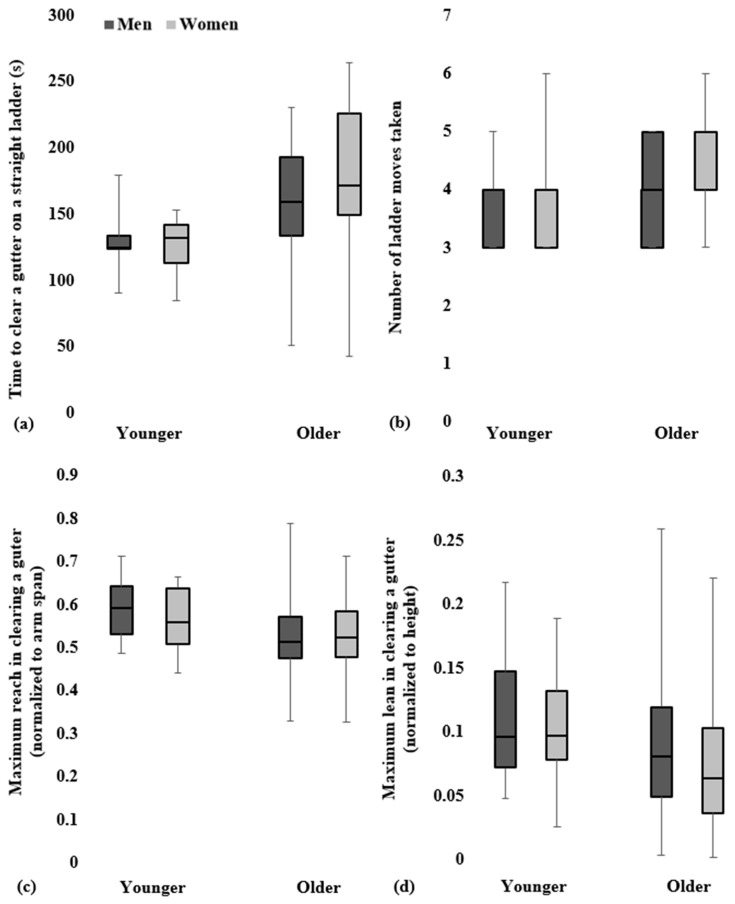
Task performance and risky ladder use metrics in clearing a simulated roof gutter on a straight ladder. Box plots of the task completion time (**a**), number of ladder moves taken (**b**), maximum reach (**c**) and maximum lean (**d**) of older and younger adults. Box plots denote the median (center line), first quartile (lower edge of box), third quartile (top edge of box), minimum (lower whisker) and maximum (upper whisker) values of the depicted groups. Depicted groups comprise men (dark gray) and women (light gray).

**Table 1 geriatrics-09-00061-t001:** Average (standard deviation) demographics for younger and older adults.

	Younger Adults	Older Adults
Age (years)	27.5 (5.2)	72.9 (5.5)
Height (m)	1.7 (0.1)	1.7 (0.1)
Weight (kg)	66.3 (13.8)	72.5 (13.8)
Technical or University Education (Years)	5.2 (1.)	5.2 (3.1)
Number of Diseases	0.3 (0.6)	2.3 (1.5)
Number of Medications	0.4 (0.8)	3.5 (3.4)
General Anxiety Disorder-7 (Score) ^+^	2.9 (3.1)	1.4 (1.9)
Physical Assessment (score)	25.3 (10.1)	35.1 (11.4)
Cognitive Processing Speed (s) ^^^	35.4 (10.3)	18.0 (6.1)
Executive Function (s) ^^^	43.2 (24.3)	22.1 (13.0)

^+^ A score of 4 or lower denotes minimal anxiety. ^^^ Cognitive processing speed and executive function were quantified from time to complete Trail Making Test A and Trail Making Test B-A, respectively.

**Table 2 geriatrics-09-00061-t002:** Median ladder exposure and mean (standard deviation) behavior metrics across age groups by gender. The {Wald Chi-Square} [df,N] (*p*-value) or F-value [df1,df2] (*p*-value) are presented for age group and gender effects. Bold values denote a statistical significance.

		Mean (Standard Deviation) or Median						Main Effects		
		Old			Young			Age Group	Gender	Age Group × Gender
		Men	Women	All	Men	Women	All			
*Ladder use frequency*										
Stepladder		Monthly	Monthly	Monthly	Few times a year	Few times a year	Few times a year	**{7.96}** **[1,122]** **(0.005)**	{2.80} [1,122] (0.094)	{1.04} [1,122] (0.308)
Straight ladder		Few times a year	Never	Once a year	Once a year	Once a year	Once a year	{0.66} [1,122] (0.417)	**{7.14}** **[1,122]** **(0.008)**	{1.42} [1,122] (0.234)
Fixed ladder		Never	Never	Never	Few times a year	Few times a year	Few times a year	**{18.31}** **[1,122]** **(<0.001)**	{1.83} [1,122] (0.176)	{0.05} [1,122] (0.817)
*Surveyed behaviors*										
Unsafe ladder behaviors ^1^		4.1 (2.6)	3.0 (2.5)	3.6 (2.6)	6.3 (3.0)	5.4 (3.9)	5.9 (3.4)	**11.48** **[1,117]** **(<0.001)**	2.18 [1,117] (0.143)	0.02 [1.117] (0.895)
Everyday risk-taking		24.5 (4.7)	24.2 (5.0)	24.4 (4.8)	31.6 (3.2)	30.0 (3.3)	30.8 (3.2)	**32.03** **[1,120]** **(<0.001)**	0.72 [1,120] (0.398)	0.31 [1,120] (0.576)
Fear of falling		13.2 (3.4)	15.0 (4.1)	14.1 (3.9)	N.A.	N.A.	N.A.	N.A.	**6.87** **[1,102]** **(0.010)**	N.A.
*Clearing a roof gutter on a straight ladder*										
Maximum reach (normalized to arm span)		0.53 (0.09)	0.52 (0.08)	0.53 (0.09)	0.59 (0.08)	0.56 (0.08)	0.58 (0.08)	**6.06 [1,117]** **(0.015)**	0.55 [1,117] (0.458)	0.39 [1,117] (0.534)
Maximum lean (normalized to height)		0.09 (0.06)	0.07 (0.05)	0.08 (0.05)	0.11 (0.05)	0.11 (0.05)	0.11 (0.05)	**4.77** **[1,109]** **(0.031)**	0.36 [1,109] (0.552)	0.03 [1,109] (0.860)
*Riskiest ladder choice* *to ^2^*										
Wash a window at home		6 (2)	5 (2)	5 (2)	N.A.	N.A.	N.A.	{1.42} [1,472] (0.234)	**{5.04}** **[1,472]** **(0.025)**	{<0.00} [1,472] (0.930)
Wash a window in the laboratory		7 (2)	6 (2)	6 (2)	7 (1)	6 (1)	7 (1)			
Avoid a 6-h wait in an abandoned warehouse		8 (1)	7 (2)	7 (1)	8 (1)	7 (1)	8 (1)			
Avoid an overnight wait in an abandoned warehouse		8 (1)	7 (1)	8 (1)	8 (1)	7 (1)	8 (1)			

^1^ Unsafe ladder behaviors—the number of unsafe ladder behaviors participants self-reported. Higher values indicate more unsafe ladder behaviors when using a ladder. ^2^ Riskiest ladder choice—each number is linked to a ladder choice. Higher values indicate a risky ladder choice (from 1 indicating the one-step block to 8 indicating the 12 ft. pole–peg ladder).

**Table 3 geriatrics-09-00061-t003:** Mean (standard deviation) ladder use ability metrics across age group by gender. The F-value [df1,df2] (*p*-value) are presented for age group and gender effects. Bold values denote a statistical significance.

	Mean (Standard Deviation)						Age and Gender Effects		
	Old			Young			Age Group	Gender	Age Group × Gender
	Men	Women	All	Men	Women	All			
*Changing a lightbulb on a household step ladder*									
Task competition time (s)									
Single task	22.8 (6.3)	25.8 (8.6)	24.3 (7.6)	17.0 (3.2)	18.8 (3.9)	17.9 (3.6)	**22.35** **[1,119]** **(<0.001)**	2.68 [1,119] (0.104)	0.05 [1,119] (0.828)
Dual task	25.3 (6.7)	28.9 (9.8)	27.1 (8.5)	18.5 (3.0)	20.0 (2.8)	19.3 (2.9)			
Elliptical area (mm^2^)									
Single task	1719 (1129)	1051 (693)	1389 (993)	1693 (877)	1226 (425)	1472 (723)	3.38 [1,111] (0.069)	**11.11 [1,111]** **(0.001)**	0.66 [1,111] (0.418)
Dual task	1814 (1037)	1085 (628)	1435 (921)	2732 (2696)	1545 (764)	2169 (2065)			
Edge distance (mm)									
Single task	72 (22)	72 (23)	72 (22)	86 (19)	92 (15)	89 (17)	**4.88 [1,111]** **(0.029)**	0.11 [1,111] (0.739)	0.52 [1,111] (0.474)
Dual task	75 (21)	73 (21)	74 (21)	76 (35)	80 (24)	78 (29)			
Animal naming rate (animals/s)									
Dual task	0.4 (0.2)	0.5 (0.2)	0.4 (0.2)	0.7 (0.1)	0.6 (0.2)	0.6 (0.2)	**23.80 [1,119]** **(<0.001)**	0.54 [1,119] (0.464)	1.10 [1,119] (0.296)
*Clearing a roof gutter on a straight ladder*									
Task completion time (s)	160.7 (40.2)	186.4 (54.7)	173.4 (49.4)	126.0 (28.9)	129.8 (21.0)	128.0 (24.4)	**19.86 [1,112]** **(<0.001)**	2.03 [1,112] (0.157)	0.56 [1,112] (0.456)
Number of ladder moves taken	4.0 (0.8)	4.4 (0.8)	4.2 (0.8)	3.4 (0.7)	3.9 (1.0)	3.7 (0.9)	6.27 [1,112] (0.014)	3.68 [1,112] (0.058)	0.09 [1,112] (0.768)

**Table 4 geriatrics-09-00061-t004:** Cognitive demand, ladder exigency scenario and interaction effects on ladder use ability and behavior metrics. The F-value [df1,df2] (*p*-value) or {Wald Chi-Square} [df,N] (*p*-value) are presented for main and interaction effects. Bold values denote a statistical significance.

Changing a Lightbulb on a Household Stepladder	Cognitive Demand	Age Group × Cognitive Demand	Gender × Cognitive Demand	Age Group × Gender × Cognitive Demand
Task completion time	**12.62 [1,119]** **(<0.001)**	0.23 [1,119] (0.632)	<0.01 [1,119] (0.967)	0.07 [1,119] (0.786)
Elliptical area	2.38 [1,111] (0.126)	1.01 [1,111] (0.318)	0.08 [1,111] (0.776)	0.08 [1,111] (0.780)
Edge distance	2.18 [1,111] (0.143)	4.97 [1,111] (0.028)	0.30 [1,111] (0.585)	0.03 [1,111] (0.872)
Ladder choice in use scenarios	Exigency scenario	Age group **×** use scenario	Gender **×** use scenario	Age group **×** gender **×** use scenario
Riskiest ladder choice	**{74.43} [3,472]** **(<0.001)**	{1.27} [2,372] (0.260)	{0.10} [3,472] (0.752)	{<0.00} [2,372] (0.955)

## Data Availability

The data presented in this study are available on request from the corresponding author.

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
