# Peer review of "Ladder Use Ability, Behavior and Exposure by Age and Gender"

_geriatrics, 2024, doi:10.3390/geriatrics9030061_

Round 1
Reviewer 1 Report
Comments and Suggestions for Authors
Baseline evaluation of enrolled subjects isn't well explained (comorbidities, drugs).
Lack of comprehensive geriatric and cognitive assessment could be a potential confounder for the older adults' group.
Author Response
We would like to thank the reviewers for their time and feedback. We believe your comments have helped increased the clarity and strength of our manuscript. Below we number each comment. Our responses our italicized and our changes in the manuscript and below are highlighted.
Reviewer 1
- Baseline evaluation of enrolled subjects isn't well explained (comorbidities, drugs).
We have added a demographics table to the manuscript to better characterize our participants.
Table 1: Average (standard deviation) demographics for younger and older adults.
|
|
Younger Adults |
Older Adults |
|
Aged (years) |
27.5 (5.2) |
72.9 (5.5) |
|
Height (m) |
1.7 (0.1) |
1.7 (0.1) |
|
Weight (kg) |
66.3 (13.8) |
72.5 (13.8) |
|
Technical or University Education (Years) |
5.2 (1.) |
5.2 (3.1) |
|
Number of Diseases |
0.3 (0.6) |
2.3 (1.5) |
|
Number of Medications |
0.4 (0.8) |
3.5 (3.4) |
|
General Anxiety Disorder-7 (Score)+ |
2.9 (3.1) |
1.4 (1.9) |
|
Grip Strength (kg) |
25.3 (10.1) |
35.1 (11.4) |
|
Cognitive Processing Speed (s)^ |
35.4 (10.3) |
18.0 (6.1) |
|
Executive Function (s)^ |
43.2 (24.3) |
22.1 (13.0) |
+a score of 4 or lower denotes minimal anxiety.
^Cognitive processing speed and executive function were quantified from time to complete the Trail-Making Test A and Trials-Making Test B-A, respectively.
- Lack of comprehensive geriatric and cognitive assessment could be a potential confounder for the older adults' group.
We agree that physical and cognitive differences between younger and older adults contribute to our age effect. After considering this comment, we determined that we could have done more to present the differences between these groups and discuss these aspects of aging in the discussion. Younger and older adults completed grip strength, cognitive processing and executive functioning assessments. Younger adults performed better in these assessments than older adults. These scores were added to the participant demographics table (see response to comment 1). Our previous work found better performance in physical and cognitive assessments to be associated with more effective ladder use in older adults (Pliner et al. 2020; 2021). This provides motivation for this study and is already stated in our introduction. Other added text in the discussion is as follows.
Line 319: Younger adults performed better than the older adults with respect to most ladder task time and stability measures. This is likely linked to greater physical and cognitive abilities in younger adults than older adults, and agrees with ladder use literature that has found greater strength and better balance and cognitive to be associated with faster ladder tasks completion times [21,22].

Reviewer 2 Report
Comments and Suggestions for Authors
This paper deals with an interesting study on ladder abilities. There aren't many fixes, but the following points should be improved.
1. The title "Effects" is inappropriate for this study. I think it should be called ``Influences'' or ``age and gender-difference.'' This is because many factors other than age and gender are assumed to have an influence.
2. I feel that it would be good to have a little more information on recruiting subjects in ``2.1 Participants'' of the research method. This study concerns how we consider the population and how far we intend to generalize our results, so we should increase the amount of information to enhance readers' understanding from that perspective.
3. It is better to mention the problem of generalization of this research in ”study limitations" starting from line 380. The scope of generalizability of this study may not be wide, including whether it applies to other ethnic groups and elderly people with high physical strength and functionality.
4. It seems that the information in document 45 is insufficient.
Author Response
We would like to thank the reviewers for their time and feedback. We believe your comments have helped increased the clarity and strength of our manuscript. Below we number each comment. Our responses our italicized and our changes in the manuscript and below are highlighted.
Review 2
This paper deals with an interesting study on ladder abilities. There aren't many fixes, but the following points should be improved.
- The title "Effects" is inappropriate for this study. I think it should be called “Influences'' or ``age and gender-difference.'' This is because many factors other than age and gender are assumed to have an influence.
We have changed our title to the following.
Ladder use ability, behavior and exposure by age and gender
- I feel that it would be good to have a little more information on recruiting subjects in ``2.1 Participants'' of the research method. This study concerns how we consider the population and how far we intend to generalize our results, so we should increase the amount of information to enhance readers' understanding from that perspective.
Additional detail was added on participant recruitment. Additional participant demographic detail was also added, see comment #1.
Line 74: One-hundred and four healthy older adults and 20 healthy younger adults were recruited through flyers in the greater Sydney area, a volunteer research registry list for Neuroscience Research Australia, and word-of-mouth.
- It is better to mention the problem of generalization of this research in “study limitations" starting from line 380. The scope of generalizability of this study may not be wide, including whether it applies to other ethnic groups and elderly people with high physical strength and functionality.
We believe the physical ability and strength of older adults in our study sample was greater than the population due to our screening methods that required participants to have some comfort level with ladder use. This may mean that our study generalizes better to older ladder users than older adults. We have acknowledged the potential limitation of generalization of these findings to other populations.
Line 400: Second, the generalization of these findings to other populations warrants investigation.
- It seems that the information in document 45 is insufficient.
Thank you for catching this error. We have updated reference, now #50.

Reviewer 3 Report
Comments and Suggestions for Authors
Many thanks to the editors for considering me to review this work and thanks to the authors for the time they have devoted to it.
The following is a series of comments in order to be able to contribute to the work.
Regarding the abstract, it is recommended that the authors make explicit reference to the instruments beyond an interview. Were more instruments used in the other experiments? It would also be necessary to refer to the type of analysis and methodology used.
As for the introduction, a more detailed analysis of the state of the art should appear, so that it can be structured on the basis of the variables or subsequent experiments. Point 2.2 is clearly and specifically explained. As for the protocols, they are also explained specifically, and it would be good if in the introduction mention were made of studies that have used these or similar processes.
The description of the statistical analysis is detailed and part of this description could be in the summary.
As for the results, they appear in detail, although it would be good to summarize them a little more and show in the tables only those results that are intended to be highlighted, this would help the reader. It is only a suggestion, it is not necessary to do so if they do not consider it necessary.
Regarding the discussion, it would be good if the authors included in their first paragraph the implications of the improvements found in their study on the lives of the subjects.
On the other hand, it would be good that the authors, in their last paragraph, indicate which are the lines of work that this study can open.
Finally, with respect to the conclusions, the authors should order them in such a way that they respond to their hypotheses and/or conclusions. It should be very clear.
Author Response
We would like to thank the reviewers for their time and feedback. We believe your comments have helped increased the clarity and strength of our manuscript. Below we number each comment. Our responses our italicized and our changes in the manuscript and below are highlighted.
Reviewer 3
Many thanks to the editors for considering me to review this work and thanks to the authors for the time they have devoted to it. The following is a series of comments in order to be able to contribute to the work.
- Regarding the abstract, it is recommended that the authors make explicit reference to the instruments beyond an interview. Were more instruments used in the other experiments? It would also be necessary to refer to the type of analysis and methodology used.
Thank you for the recommendation to further explain our experiments. Due to the word limit of the abstract, we are not able to fully explain our instruments and methods. We have added the following text to provide additional clarity.
Line 14: Ladder use ability and behavior data was captured from recorded time, performance, motion capture and user choice data.
- As for the introduction, a more detailed analysis of the state of the art should appear, so that it can be structured on the basis of the variables or subsequent experiments. Point 2.2 is clearly and specifically explained. As for the protocols, they are also explained specifically, and it would be good if in the introduction mention were made of studies that have used these or similar processes.
We have added an additional paragraph to the introduction that highlights that gaps in experimental ladder safety research to further support our aims.
Line 52: Experimental ladder safety research has focused primarily on a single age group [16,17,21,22,28-38]. The majority of this research has been conducted on healthy younger adults or occupational ladder users [16,17,28,29,31,33-38]. Ladder use is a complex task, requiring physical and cognitive attributes [21,22]. While age is known to affect physical and cognitive ability, there is limited investigation on ladder use ability and behavior between age groups. Similarly, there is a needed investigation of ladder use ability and behavior between men and women.
- The description of the statistical analysis is detailed and part of this description could be in the summary.
Please respectfully disagree the need to repeat the use of statistical testing when summarizing our results.
- As for the results, they appear in detail, although it would be good to summarize them a little more and show in the tables only those results that are intended to be highlighted, this would help the reader. It is only a suggestion, it is not necessary to do so if they do not consider it necessary.
We have added a table to better summarize the population of this study. See comment #1 (Review 1). We believe it is important to show all results in the tables, even the results we choose not to highlight for reader transparency.
- Regarding the discussion, it would be good if the authors included in their first paragraph the implications of the improvements found in their study on the lives of the subjects.
We have added the following study implications to our first paragraph.
Line 317: This knowledge can guide population-targeted training programs for more effective ladder fall intervention programs.
- On the other hand, it would be good that the authors, in their last paragraph, indicate which are the lines of work that this study can open.
We have added the following sentence to our last paragraph in the discussion to better guide the reader to future lines of work.
Line 411: Overall, there is need for future studies that consider greater population variability, causal factors of ladder use behavior, and real-world scenarios.
- Finally, with respect to the conclusions, the authors should order them in such a way that they respond to their hypotheses and/or conclusions. It should be very clear.
We have added the following text to the conclusions to improve clarity around our hypothesis tested outcome.
Line 414: The aim of this study was to determine age and gender differences between ladder use ability, behavior and exposure. We (i) confirmed older adults to display poorer ladder use ability than younger adults. We (ii) did not confirm women to display poorer ladder use ability compared to men. We (iii) confirmed older adults to display less risky ladder use behaviors than younger adults; and men to choose riskier ladders to climb than women, but men and women to display similar risk during ladder use.

Reviewer 4 Report
Comments and Suggestions for Authors
Thank you for your work and for the opportunity to review this paper.
Congratulations to the authors, the article is interesting and has a good presentation and methodology.
Author Response
Thank you!
